# Integrated Analysis of microRNA and RNA-Seq Reveals Phenolic Acid Secretion Metabolism in Continuous Cropping of *Polygonatum odoratum*

**DOI:** 10.3390/plants12040943

**Published:** 2023-02-19

**Authors:** Yan Wang, Kaitai Liu, Yunyun Zhou, Yong Chen, Chenzhong Jin, Yihong Hu

**Affiliations:** 1College of Agriculture and Biotechnology, Hunan University of Humanities, Science and Technology, Loudi 417000, China; 2Hunan Province Key Laboratory of Plant Functional Genomics and Developmental Regulation, State Key Laboratory of Chemo/Biosensing and Chemometrics, National Center of Technology Innovation for Saline-Alkali Tolerant Rice, College of Biology, Hunan University, Changsha 410082, China

**Keywords:** *Polygonatum odoratum*, root rot, soil phenolics, root exudates, root, microRNA

## Abstract

*Polygonatum odoratum* (Mill.) Druce is an essential Chinese herb, but continuous cropping (CC) often results in a serious root rot disease, reducing the yield and quality. Phenolic acids, released through plant root exudation, are typical autotoxic substances that easily cause root rot in CC. To better understand the phenolic acid biosynthesis of *P. odoratum* roots in response to CC, this study performed a combined microRNA (miRNA)-seq and RNA-seq analysis. The phenolic acid contents of the first cropping (FC) soil and CC soil were determined by HPLC analysis. The results showed that CC soils contained significantly higher levels of *p*-coumaric acid, phenylacetate, and caffeic acid than FC soil, except for cinnamic acid and sinapic acid. Transcriptome identification and miRNA sequencing revealed 15,788 differentially expressed genes (DEGs) and 142 differentially expressed miRNAs (DEMs) in roots from FC and CC plants. Among them, 28 DEGs and eight DEMs were involved in phenolic acid biosynthesis. Meanwhile, comparative transcriptome and microRNA-seq analysis demonstrated that eight miRNAs corresponding to five target DEGs related to phenolic acid synthesis were screened. Among them, ath-miR172a, ath-miR172c, novel_130, sbi-miR172f, and tcc-miR172d contributed to phenylalanine synthesis. Osa-miR528-5p and mtr-miR2673a were key miRNAs that regulate syringyl lignin biosynthesis. Nta-miR156f was closely related to the shikimate pathway. These results indicated that the key DEGs and DEMs involved in phenolic acid anabolism might play vital roles in phenolic acid secretion from roots of *P. odoratum* under the CC system. As a result of the study, we may have a better understanding of phenolic acid biosynthesis during CC of roots of *P. odoratum*.

## 1. Introduction

*Polygonatum odoratum* (Mill.) Druce, a traditional Chinese herb, is one of the rhizome plants in the Liliaceous family and has widespread medicinal functions. It provides multiple pharmacological effects on the blood, blood lipid, immune, endocrine, cardiovascular, and nervous systems [1,2]. In China, *P. odoratum* is widely distributed in 12 provinces, including the northeast, northwest, southeast, and central parts [3]. However, *P. odoratum* is also susceptible to replanting diseases, which result in a severe decline in the biomass and quality of underground tubers to continuous cropping (CC). Furthermore, numerous studies have reported that most medicinal plants were attacked by replanting diseases, which negatively impacts their growth and development, and they are even harvestless [4,5]. For example, ginseng yield and quality of *Panax ginseng* Meyer (Araliaceae) [5], *Rehmannia glutinosa Libosch* [6], and *Radix pseudostellariae L* [7] are seriously compromised under consecutive monoculture. The consecutive monoculture problem (CPM) is a major limiting factor in medicinal plants’ growth and productivity. Previous studies indicated that three major factors contribute to the CMP, including imbalance of soil microbial community [8], autotoxic substances in rhizosphere soil [9], and deterioration of soil physicochemical characteristics [10]. It is likely that *P. odoratum* will be influenced by the interactions between root exudation, soil properties, microbiome activities, and pathogen dynamics under consecutive monocultures.

Root exudates are perceived as communication signals between roots and microorganisms [11,12]. As one of the typical root exudates in Chinese medicinal plants, including *P. odoratum* [13] and *Cyclocarya paliurus* [14], phenolic acids are prone to changing the soil microbial community and indirectly causing autotoxicity in monoculture plants. Previous results have also shown that the CC of Malus domestica “M26” plants can increase the levels of phenolic acid in the root cells, acts as an antioxidant, and inhibits the growth of infectious microorganisms directly [15]. Moreover, our previous studies demonstrated that CC increased the anabolism of phenolic acid in *P. odoratum,* resulting in phenolic acids accumulation in the soil, thereby promoting the development and growth of soil-borne pathogens, inhibiting seedling growth and reducing the yield potential [13,16]. The CC of *R. glutinosa* also alters the rhizosphere’s microbial community, favoring pathogenic and toxin-producing bacteria s over beneficial microorganisms, which are mediated by the root exudates, including phenolic acids [6,17]. The soil may accumulate phenolic acids due to CC changing membrane permeability, inhibiting nutrient uptake, and inactivating endogenous plant hormones to interfere with the normal physiological process [18]. Phenolic acids, one of the important root exudates, are involved in allelopathy, alter soil microbial community, and indirectly cause autotoxicity in monoculture plants. Little still is known about the multifaceted regulatory mechanism controlling phenolic acids biosynthesis beyond transcriptional regulation in *P. odoratum* monoculture rhizospheres.

MicroRNAs (miRNAs) are a class of endogenous, small, and noncoding RNAs involved in post-transcriptional gene repression. In previous studies, miRNAs have been shown to play an important role in many biochemical reactions, including plant growth, development, and stress resistance [19,20]. In *S. miltiorrhiza*, five miRNAs and seven target genes were implicated in replanting disease in consecutive cropping plant roots [21]. Li et al. [22] have described that CC upregulated 21 miRNAs, indicating that miRNA responded to continuous monoculture, improved root growth and development, enhanced transport activity, and strengthened its tolerance to various stresses, thereby improving the productivity of *A. bidentata* as observed in the replanting benefit. Based on the micoRNA sequencing of the CC and the first cropping (FC) *P. odoratum*, we have discovered previously that miRNAs-regulated genes involved in sugar and phenylpropanoid metabolism are enriched [22]. All these results indicated that miRNA might regulate phenolic acid metabolism and involve the plant in root rot. Until now, little has been known about miRNAs in the phenolic acids biosynthesis process of the rhizosphere of *P. odoratum* under the CC system.

To systematically investigate the regulatory network between miRNAs and mRNAs associated with phenolic acid biosynthesis under the CC system, miRNAs and mRNA sequences were analyzed during phenolic acid accumulation. Using high-throughput sequencing and bioinformatics tools, we identified conserved and novel miRNAs and their potential targets. Our study found some miRNA-mRNA regulatory modules involved in the biosynthesis of phenolic acids. Under the CC system, our results may provide new insights into miRNA regulation of phenolic acids biosynthesis in *P. odoratum*.

## 2. Results

### 2.1. Rhizosphere Soil Phenolic Acids Contents during the Replanting of P. odoratum

Previously, we detected five phenolic acids in the FC and CC rhizosphere soil, where p-hydroxybenzoic, syringic acid, cumaric acid, and ferulic acid contents in the CC soil were significantly higher [13]. In addition, other phenolic acids were also identified in this study, including cinnamic acid, *p*-coumaric acid, sinapic acid, phenylacetate, and caffeic acid. Particularly, a higher concentration (*p* < 0.01) of *p*-coumaric acid, phenylacetate, and caffeic acid was found in the CC soil than those in the CF soil (Figure 1). Cinnamic acid and sinapic acid in the CC and FC soils did not differ significantly, but they tended to be higher in the CC soil than in the CF soil (Figure 1). The results showed that the phenolic acid contents in the CC soil had a higher phenolic acid content than in FC soil, according to the results.

### 2.2. Analysis of DEGs Related to Phenolic Acid Metabolism

We performed high-throughput sequencing to determine gene expression profiles in *P. odoratum* roots in response to CMP. We searched for homologous sequences using BLASTX against NR, NT, KO, SwissProt, PFAM, GO, and KOG databases. The results showed that 68.337% could be assigned to a homolog in all five databases (Figure 2A). According to the E-value distribution, we found that approximately 19.6% of the unigenes displayed very strong homology (E-value < 1.0 × 10^−100^) with available plant sequences (Figure 2B). As shown in Figure 2C, approximately 173,367 unigenes were associated with five top-hit species, including *Elaeis guineensis*, *Phoenix dactylifera*, *Musa acuminate*, *Vitis vinifera*, and *Nelumbo nucifera*.

In root tissues of *P. odoratum*, 15,788 DEGs were screened out, including 4843 upregulated DEGs and 10,945 downregulated DEGs (padj < 0.05, and the |log2ratio| ≥ 1; Figure 3A,B). Our previous study showed that the upregulated DEGs were mainly enriched in “phenylalanine,” “tyrosine and tryptophan biosynthesis,” “phenylalanine metabolism,” “phenylpropanoid biosynthesis,” and “nitrogen metabolism” [13]. In contrast, DEGs downregulated in the pathways were significantly enriched, such as “plant hormone signal transduction,” “DNA replication,” “brassinosteroid biosynthesis,” etc., and these enriched DEGs are *FBP*, *PGD*, *RPE*, *GPD*, *SORD*, *MAlZ*, *GLGC*, *GPI*, *E3.2.1.4*, *AROK*, *AROC*, *TYDC*, and *E1.10.3.1*, involved in regulating the synthesis of phenolic acids [13]. In addition to the above DEGs, we found others related to the process, such as *CAD*, *CCR*, *E1.11.1.7*, *COMT*, *CCR*, *CSE*, *AMIE*, *CYP98A*, *TYRAAT*, *ALDO*, *4CL*, *AOC3*, *HCT*, *ASP5*, *ADT*, and *SCRK* (Appendix A). These genes that encode the phenolic acid-synthesis enzymes were significantly up-regulated (*CSE*, *AMIE*, *CYP98A*, *TYRAAT*, *ALDO*, *4CL*, *AOC3*, *HCT*, *ASP5*, *ADT*, *PGD*, and *SCRK*) in CCR, while those unfavorable for the phenolic acid synthesis were down-regulated (*CAD*, *CCR*, *E1.11.1.7*, and *COMT*, Appendix A). These results indicated that these DEGs might be related to regulating phenolic acid synthesis in the CC of *P. odoratum*.

### 2.3. Differential Expression Profiles of miRNAs and Functional Analysis during the Replanting of P. odoratum

To compare the different miRNA expression profiles in the CC and FCR of *P. odoratum*, differential expression analysis of the miRNAs was performed on the normalized read count for each identified miRNA. In *P. odoratum* roots, 253 conserved and 79 novel miRNAs were identified (Appendix A). Among them, 142 DEMs were defined by comparing the TPM expression value in CC vs FC (72 down-regulated, 70 up-regulated) *P. odoratum* roots (Appendix A and Figure 4A,B). Novel_6, novel_8, osa-miR167d-5p, mtr-miR319a-3p, ptc-miR396f, novel_9, ath-miR396a-5p, gma-miR396h, and zma-miR396g-3p were the most expressed DEMs in each sample (Figure 4C).

### 2.4. Identification of the Key miRNA-mRNA Pairs Related to Phenolic Acid Synthesis during the Replanting of P. odoratum

To identify phenolic acid synthesis-associated miRNAs and mRNA interactions and understand their potential regulatory mechanisms in consecutive monoculture root tissues of *P. odoratum*, we performed association analyses between DEMs and target mRNAs related to phenolic acid synthesis. Among 641 target genes, five related to phenolic acid synthesis were screened, corresponding to eight DEMs (Figure 5 and Appendix A). Some miRNAs were upregulated, with the expression levels of their target genes decreased in CC vs. FC root tissues of *P. odoratum* (Figure 5 and Appendix A). Ath-miR172a, ath-miR172c, novel_130, and tcc-miR172d which targeted ADT (Cluster-60288.94981) were up-regulated, and sbi-miR172f was down-regulated. Further, the ADT target gene had a higher expression level in CC vs FC root tissues of *P. odoratum* (Figure 5), which is beneficial for phenylalanine synthesis. Moreover, osa-miR528-5p and mtr-miR2673a were down-regulated, whereas their common target gene [*EC:1.11.1.7*] (Cluster-60288.231604 and Cluster-60288.146690) was down-regulated, which could inhibit syringyl lignin formation [23]. Mtr-miR2673a, which targets a [*EC:1.11.1.7*] (Cluster-60288.201647), was down-regulated, while its other target AOC3 (Cluster-60288.117692, Cluster-60288.226755, Cluster-60288.146690) has a high expression level, coding for a critical enzyme involved in phenylacetate synthesis. Furthermore, a negative correlation was also found between nta-miR156f target to MALZ (Cluster-60288.201647), which has been related to the shikimate pathway (Figure 5). These results indicate that these miRNAs may be involved in the phenolic acid synthesis.

### 2.5. Regulatory Network of Phenolic Acid Biosynthesis

Based on RNA-seq data and the phenol propane biosynthesis method, we constructed the phenolic acid synthesis pathway of *P. odoratum* (Figure 6). Combining the results of all genes and miRNAs with building a network showed a more direct relationship between miRNA-regulated gene expression, gene expression, and phenolic acid accumulation. 

Phenolic acid synthesis in plants is provided mainly through the shikimate and phenylalanine pathways [24]. Among 15,788 DEGs, 220 DEGs related to phenolic acid biosynthesis were screened out, corresponding to eight DEMs. Among the DEGs, eight shikimate synthesis genes (genes encoding *MALZ*, *SCRK*, *ALDO*, *FBP*, *SORD*, *REP*, *GLGC*, and *PGD*), two chorismate synthesis genes (genes encoding *AROK*, and *AROC*), and eleven phenolic acid synthesis genes (genes encoding *ADT*, *AOC3*, *CYP73A*, *CYP98A*, *4CL*, *HCT*, *CSE*, *COMT*, *CAD*, *EC:1.11.1.7*, and *AMIE*), such as phenylacetate, signapic acid, caffeic acid, and *p*-coumaric acid, etc., were screened in the roots of *P. odoratum* (Figure 6 and Appendix A). Moreover, the eight DEMs, in turn, regulated their target genes and thus synthesized phenolic acid. miR156f targeted malz, tcc-miR172d, sbi-miR172f, novel_130, ath-miR172a, and ath-miR172c targeted ADT, mtr-miR2673a targeted *E3.1.1.11*, *EC:1.11.1.7* and *AOC3*, and osa-miR528-5p targeted *E1.11.1.7*. Almost all target genes were beneficial to synthesizing phenolic acids (Figure 6). Based on the above results, these miRNAs may also play a key role in phenolic acid biosynthesis.

### 2.6. Quantitative PCR (qPCR) Validation of the Key mRNA and miRNA Related to Phenolic Acid Synthesis

As a further validation of our RNA-seq results, we selected 19 unigenes for qPCR analysis. By utilizing shikimic acid, plants produce phenolic acids via the phenylpropanoid pathway. Hence, shikimic acid is the first step in the biosynthesis of phenolic acids in plants [24]. Among them, seven shikimic acid synthesis-related enzymes, namely *MALZ* (60288.201647), *ALDO* (60288.23879), FBP (60288.235250), *SORD* (60288.223504), *REP* (60288.218462), *PGD* (60288.211711), and *GLGC* (60288.33753), were selected for the qPCR. CCR compared with the CFR significantly up-regulated the expression of the other shikimic acid synthesis enzyme genes except for *GLGC* (Figure 7A), whereas that of the degrading genes (*GPI* (60288.243910) and *E3.2.1.4*: endoglucanase (60288.10931)) decreased (Figure 7A). The synthesis of phenolic acids in plants mainly involves the deamidation of phenylalanine. To a lesser degree, tyrosine produces cinnamic and/or *p*-coumaric acids. Following hydroxylation and methylation, cinnamic and *p*-coumaric acids form ferulic and caffeic acids [25,26]. *AROK* (60288.274987) and *AROC* (60288.232788), which encode enzymes that catalyze shikimate to phenylalanine and tyrosine, were up-regulated in CCR compared to CFR (Figure 7A). Similarly, phenylacetate biosynthesis (*AOC3* (60288.225005), *AMIE* (60288.184064)), *p*-coumaric acid (*CYP73A*, 60288.172246), caffeoyl shikimic acid (*CYP98A*, 60288.232186), *4CL* (60288.271800)), caffec acid (*CSE*, 60288.208625), and tyramine (*TYRAAT* (60288.107598), *TYDC* (60288.348666)) biosynthesis-related genes were up-regulated (Figure 7A), whereas the lignin biosynthesis genes (*E1.11.1.7*, 60288.172775) decreased (Figure 7A), which indicated that CCR was more conducive to phenolic acid synthesis, and a Pearson correlation coefficient of 0.4845 confirmed the reliability of the transcriptome sequencing results (Figure 7C).

To validate the consistency of the miRNA and sRNA sequencing results, eight miRNAs were randomly selected for qRT-PCR analysis (Figure 7B). The results showed that the relative expression levels of eight miRNAs correspond with those obtained from miRNA sequencing results (Figure 7B). A significant Pearson correlation coefficient of r^2^ = 0.772 indicates the reliability of by sRNA-Seq data (Figure 7D).

## 3. Discussion

CMP, also known as replanting disease, is common to Chinese medicinal herbs in all growing regions, leading to a severe reduction in biomass and crop quality. However, the replanting disease has been attributed to many factors, including soil nutrient imbalance, physical and chemical properties, accumulation of autotoxins generated by roots, and changes in the soil microbial community structure [27,28]. Among them, the soil phenolic acids generated by roots represent an important contribution to replant disease. In addition to acidifying soil, phenolic acids induce phytotoxicity and affect soil enzyme activity, nutrient cycling, and ion uptake, thereby inhibiting the plant’s growth [29,30]. Continuous cropping of strawberries, either hydroponically or with solid medium, accumulated four phenolic acids, namely *p*-hydroxybenzoic, ferulic, cinnamic, and *p*-coumaric acids [31,32]. Similarly, we previously detected five phenolic acids in the FC and CC rhizosphere soil, with significantly higher levels found in the CC soil for *p*-hydroxybenzoic, syringic, cumaric, and ferulic acid [13]. In the present study, we identified five other rhizosphere soil phenolic acids, three differentially accumulated in the CC and FC soil, i.e., *p*-coumaric acid, phenylacetate, and caffeic acid (Figure 1). These results indicated that the CC system increased phenolic acid accumulation from the rhizosphere soil of *P. odoratum*. Phenolic acids, as allelopathic substances, are mainly secreted by plant rhizomes but can also be decomposed by old roots and produced by soil microorganisms [18,33]. In the present study, we speculated that they might be secreted by the rhizomes of *P. odoratum*, thereby contributing to CMP and replant disease.

Plant phenolic acids are mainly derived via the phenylpropanoid and tyrosine pathways using shikimic acid [34]. There are three major pathways for making metabolic intermediates in plants: pentose phosphate, starch, and sucrose metabolism and pentose and glucuronate interconversions, which can provide erythrose-4-phosphate as a precursor to the *synthesis* of shikimic acid. In the shikimic acid biosynthesis pathway, *FBP*, *PGD*, *SORD*, *RPE*, *GPI*, *MALZ*, and *GLGC* are essential enzymes that produce shikimic acid [35,36,37,38]. Our previous study revealed that the pentose phosphate pathway-related genes (*FBP* and *PGD*), starch and sucrose metabolism-related gene (*MALZ*), and pentose and glucuronate interconversions related gene (*SORD*) during CC of roots of *P. odoratum* were significantly upregulated, which was benefic to the shikimic acid formation [13]. In this study, we found that *ALDO* encoded fructose-bisphosphate aldolase and was able to regulate the fructose-6-phosphate synthesis, which was used to synthesize shikimic acid (Figure 6). Shikimic acid is the precursor of chorismate via shikimate kinase (AROK) and chorismate synthetase (*AROC*) in the shikimate pathway [39]. In contrast, phenolic acids are formed downstream of the chorismate, starting with the phenylpropanoid and tyrosine pathways [34]. As a precursor of phenolic acids, *AROK*, *AROC*, *ADL*, *4CL*, *CYP73A*, *AOC3*, *AMIE*, and *COMT* are critical enzymes to produce various phenolic acids [40,41]. In this study, except for *COMT*, the other seven genes were markedly upregulated in the CCR (Figure 6). As a precursor of phenolic acids, *AROK* encoded shikimate kinase and *AROC* chorismate synthase. *ADT* and *AOC3* encoded arogenate dehydratase and primary-amine oxidase, which promote phenylacetate biosynthesis. *CYP73A* encoded trans-cinnamate 4-monooxygenase, which converts cinnamic acid to *p*-coumaric acid through hydroxylation, whereas the CoA of *p*-coumaric acid is esterified by 4-Couramarate CoA (4CL) [42] and then converted to caffeoyl shikimic acid or caffeic acid in the presence of shikimate O-hydroxycinnamoyl transferase (HCT) or caffeoyl shikimate esterase (CSE) (Figure 6), which is supportive of our results that *p*-coumaric acid, phenylacetate, and caffeic acid were significantly increased in the CC soil (Figure 1). Conversely, the downregulation of *COMT* and *CAD*, which encoded caffeic acid 3-O-methyltransferase and cinnamyl-alcohol dehydrogenase, could inhibit lignin biosynthesis [43]. The *p*-coumaric acid, in turn, could be synthesized from more materials in this process. These findings further supported that the *P. odoratum* in the CC system might result in an accumulation of phenolic acids secreted and suggested that these genes may play essential roles in all examined phenolic acids accumulation. Another pathway for phenolic acid synthesis is mainly derived from the tyrosine pathway. Tyrosine is synthesized from chorismate by arogenate dehydrogenase (TYRAAT), which is then converted to tyramine through Tyrosine decarboxylase (TYDC) [44]. Plants can conjugate partial phenolic acids with tyramine, especially under stress [45], which activates the shikimate, phenylpropanoid, and arylmonoamine pathways [13,46], thereby accumulating more phenolic acids. Our previous study found that *TYDC* in CCR was significantly upregulated. In addition, we also found that *TYRAAT* was upregulated in this study, further suggesting that the continuous cropping of *P. odoratum* promoted the anabolism and accumulation of phenolic acid.

Mounting evidence suggests that miRNAs play an essential role in regulating target genes under adverse stress. We markedly identified 142 DEMs through psRobot_tar in the CC vs FC root tissues, suggesting that these miRNAs in the *P. odoratum* were involved in the CMP, and CMP was an extremely complex biological process. Most studies have shown that miRNAs bind to target genes’ 30 untranslated regions (UTR) to suppress their expression [47]. However, in addition to sbi-miR172f overexpression down-regulating *ADT*, ath-miR172a, ath-miR172c, novel_130, and tcc-miR172d overexpression upregulated *ADT*. We inferred that after ath-miR172a, ath-miR172c, novel_130, and tcc-miR172d were overexpressed, *ADT* was upregulated in a compensatory manner. miRNA can act as positive regulators of target genes expression and positively regulate gene transcription by binding to targeted promoters, a process known as RNA activation [48]. This may be because a subset of th-miR172a, ath-miR172c, novel_130, and tcc-miR172d can activate *ADT* transcription from enhancer sites and function as an activator [49]. ADT is an arogenate dehydratase enzyme that catalyzes L-arogenate or prephenate covert to phenylalanine [50], thus increasing the phenolic acid contents of root exudates. These results implied that ath-miR172a, ath-miR172c, novel_130, tcc-miR172d, and sbi-miR172f may relate to the synthesis of phenolic acids and the increase of the CMP of the *P. odoratum* (Figure 6). On the contrary, osa-miR528-5p and mtr-miR2673a down-regulated their target (*EC:1.11.1.7*) and inhibited syringyl lignin formation [23], which could be favorable for phenolic acid accumulation. Another target of mtr-miR2673a is *AOC3,* which codes for a critical enzyme involved in phenylacetate synthesis, which can regulate phenolic acid synthesis in the shikimate pathway (Figure 6). These results suggested that miRNAs may also play crucial roles in phenolic acid biosynthesis and accumulate in continuous cropping.

## 4. Materials and Methods

### 4.1. Plant Materials

The roots and rhizosphere soil of *P. odoratum* were kindly supplied by the research group of Yihong Hu [13]. The CC refers to *P. odoratum* grown in the land where the same plants were harvested. FC denotes *P. odoratum* planted the fields near CC where the cabbages had been harvested.

Following our previous method [13], rhizosphere soil was collected and preprocessed. After gentle shaking, rhizosphere soil adheres to the roots in total g. After shaking off loosely adhering soil, the rhizosphere soil was collected by brushing its circumference (0 to 5 mm), air-dried at room temperature, passed through the 2 mm mesh sieve, and stored at 4 °C until soil phenolic acid was analyzed. During the rhizome expansion stage, root samples of consecutive cropping plants (CCR) and first cropping plants (FCR) were collected randomly from the fields for microRNA and RNA-seq analysis. After collection, all samples were immediately frozen in liquid nitrogen and stored at −80 °C until further investigation.

### 4.2. Quantification Analysis of Phenolic Acid from Rhizosphere Soil

Rhizosphere soil phenolic acid was determined according to our previous description [13]. In brief, the rhizosphere soil was extracted using NaOH, adjusted to pH 2.5 with HCl, and extracted twice with ethyl acetate. The pooled extracts were concentrated to dryness, dissolved in methanol, and filtered through a 0.45-μm membrane. Quantification analysis of phenolic acids from rhizosphere soil was carried out using a Rigol high-performance liquid chromatography (HPLC) L300 (Rigol Technologies, Beijing, China) equipped with a Kromasil C18 column (Akzo Nobel, Amsterdam, The Netherlands). At a constant flow of 1 mL/min, 1% phosphoric acid (eluent A) and 100% methanol (eluent B) were used in a gradient. Finally, based on the retention times and adding pure standards to the sample, phenolic acids were identified and quantified at 280 nm.

### 4.3. RNA Isolation, Quantification, and Qualification

TRIzol reagent (Invitrogen, Carlsbad, CA, USA) was used to isolate total RNA from root tissues, which was treated with RNase-free DNase I to remove contaminants. A nanophotometer (Implen Inc., Westlake Village, CA, USA) was used to measure purity, a Qubit RNA Assay Kit (Life Technologies, Carlsbad, CA, USA) to assess concentration, and a Nano 6000 kit (Agilent Technologies, Santa Clara, CA, USA) to estimate integrity. The total RNA was immediately used for mRNA library construction, miRNA library preparation in Novogene Technology Co., Ltd. (Beijing, China), and quantitative real-time PCR (qRT-PCR) verification.

### 4.4. RNA-Sequencing Analysis

The FC and CC *P. odoratum* roots transcriptome was analyzed using two libraries (CCR and FCR) designed for RNA-Seq in our previous research [13]. Novogene Technology Co., Ltd. (Beijing, China) prepared and sequenced the library. Illumina HiSeq 2500 was used to sequence each library using paired-end protocols of 150 bp. The raw pair-end was used as clean reads.

To obtain clean reads from mRNA sequencing data, adapters, low-quality reads, and ambiguous were removed. Then, Trinity (version 2.0.6) was used for the de novo transcriptome assembly [51]. Each cluster’s longest sequence region was selected as a unigene. Annotation of the assembled unigenes was performed on NR (NCBI non-redundant protein sequences database), Nt (NCBI nucleotide sequences database), Pfam (Protein family database), KOG (eukaryotic orthologous groups), COG (Clusters of Orthologous Groups), SwissProt (Swiss Institute of Bioinformatics databases), and KEGG (Kyoto Encyclopedia of Genes and Genomes databases). As a normalization method, reads per kb per million reads (RPKM) were calculated for each unigene. With the gene symbol annotation, the differential expression (*n* = 3) was screened using DESeq R package (Version 1.10.1) (|log2ratio| ≥ 1 and padj < 0.05) [52]. Then, the DEGs were subjected to KEGG pathway annotation using KOBAS (version 2.0). Pathways with a false discovery rate (FDR) ≤ 0.05 were significantly more enriched in DEGs [53].

### 4.5. Bioinformatics Analysis of Small RNA Sequences

The sequences of 18–30 nt clean reads were obtained by removing 5′ adapter-contaminated reads, low-quality reads, and ambiguous reads for small RNA analysis [54]. The clean reads were then mapped against miRbase 20.0 databases (http://www.mirbase.org (10 August 2017)) [53]. The Rfam (Version 13.0) database was used to filter the rRNA, tRNA, snRNA, and snoRNA [55]. By comparing the sRNA reads with the known plant miRNAs in the miRBase 20.0, the conserved miRNAs were identified. The unannotated reads were used to predict novel miRNAs using miREvo [56]. 

miRNAs’ expression was calculated and normalized using TPM (transcripts per kilobase million) [57]. Based on the fold difference in the expression level (|log2 fold change| ≥ 1) and the significance of the expression difference (*p*-value < 0.05), differentially expressed miRNAs (DEMs) in roots (FCR and CCR) were analyzed using DESeq [58]. Finally, we predicted the target genes for DEMs sequences using psRNA target [59]. Based on the corresponding genes of deferentially expressed miRNA, GO and KEGG enrichment analyses were performed with GOseq [60] and KOBAS 2.0 [57].

### 4.6. Quantitative Real-Time PCR Validation

To verify phenolic acid biosynthesis genes as well as miRNA, twenty genes and eight miRNAs were selected for quantitative real-time PCR (qRT-PCR) verification according to our previous methods [13]. cDNA was synthesized by a PrimerScript RT Mix (Takara, Beijing, China) using total RNA for mRNA and miRNA library construction to validate the mRNA expression level. Similarly, miRNA expression levels were validated by reverse-transcribing 1 µg of the above total RNA into cDNA using miR-X miRNA First-Strand Synthesis Kit (Takara, Beijing, China). Finally, the CFX Connect Real-Time PCR Detection System (Bio-Rad, Hercules, CA, USA) was used to perform the qRT-PCR analysis using the TB Green Premix Ex TaqII (Takara, Beijing, China). We use the R = 2^^−ΔΔCt^ method to calculate each mRNA’s and miRNA’s relative expression levels [61]. 18S and U6 genes were used as the internal reference gene. Three technical replicates per sample were performed. Appendix A lists all primers used in this test.

### 4.7. Statistical Analysis

Data were expressed as the mean ± standard error of the mean (SEM). Statistical significance was assessed by the independent sample *t*-test. SPSS (SPSS version 21.0 for Windows; SPSS Inc., Chicago, IL, USA) software packages were used to statistical significance. There were significant differences when *p* < 0.05, and a tendency when 0.05 < *p* < 0.1. Graphs were made using GraphPad Prism version 5.01 (GraphPad Software, La Jolla, CA, USA).

## 5. Conclusions

In conclusion, in the present study, we identified five other rhizosphere soil phenolic acids, three of which were differentially accumulated in the CC and FC soil, i.e., *p*-coumaric acid, phenylacetate, and caffeic acid. The genomic and miRNA analysis shows for the first time the miRNA-mRNA regulatory network that may exist in phenolic acids biosynthesis. We hope that the phenolic acids biosynthesis mechanism in the continuous cropping of *P. odoratum* can be further improved. The results of this study will contribute to further understanding the biosynthesis of phenolic acids in the CC of roots of *P. odoratum*.

## Figures and Tables

**Figure 1 plants-12-00943-f001:**
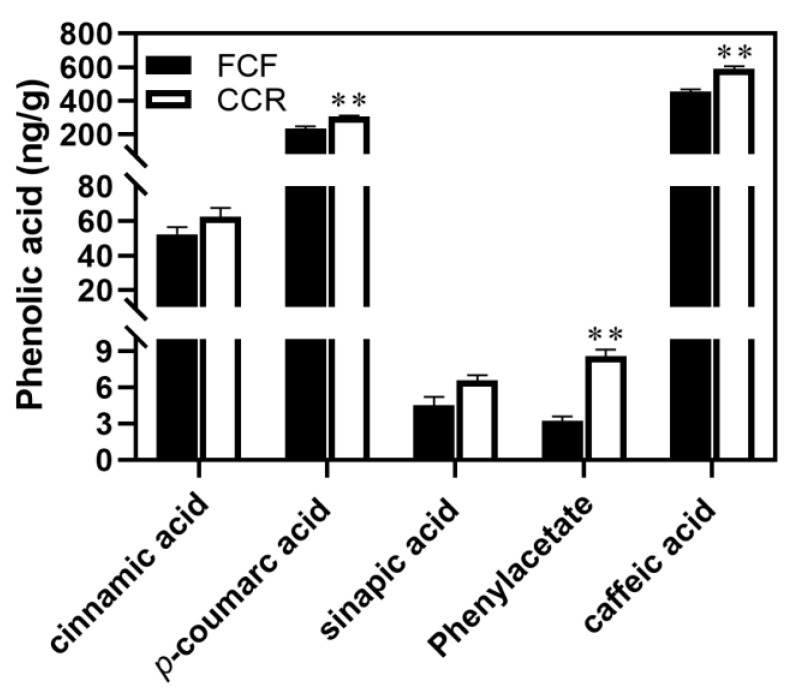
Content of metabolites in phenolic acid biosynthesis pathway in FC and CC rhizosphere soil. ** represent significant at *p* ≤ 0.01 levels according to Student’s t-test, respectively.

**Figure 2 plants-12-00943-f002:**
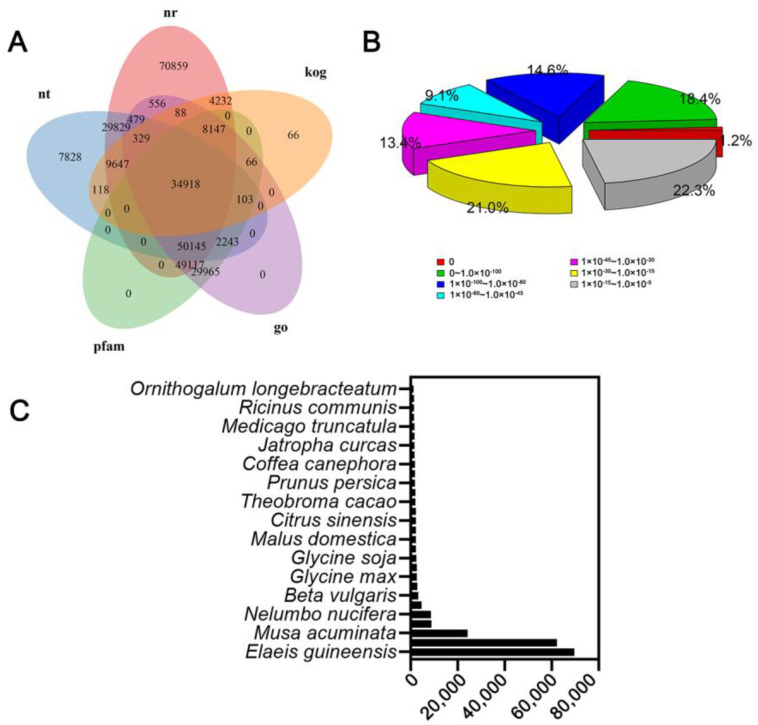
Homology search characteristics of P. odoratum unigenes. (**A**) A Venn diagram showing the number of unigenes annotated by BLASTx. A circle indicates the number of unigenes annotated by single or multiple databases. (**B**) E-value distribution of the top BLASTx hits against the Nr database. (**C**) The number of unigenes in Nr database that match the 30 top species using BLASTx.

**Figure 3 plants-12-00943-f003:**
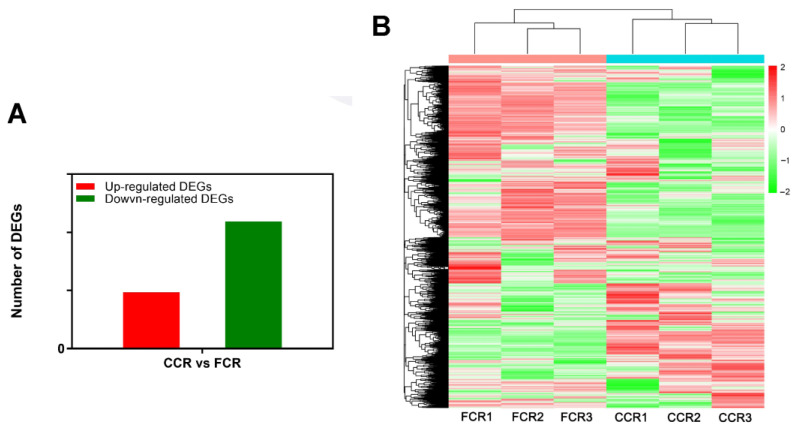
Analysis of differential gene expression in CC vs FC root tissues of *P. odoratum*. (**A**) Up-regulated and down-regulated the amount of DEGs. FCC stands for the root samples of the first cropping, and CCR stands for root samples of consecutive cropping. (**B**) Heatmap and clustering analysis of DEGs. FCC stands for the root samples of the first cropping, and CCR stands for root samples of consecutive cropping.

**Figure 4 plants-12-00943-f004:**
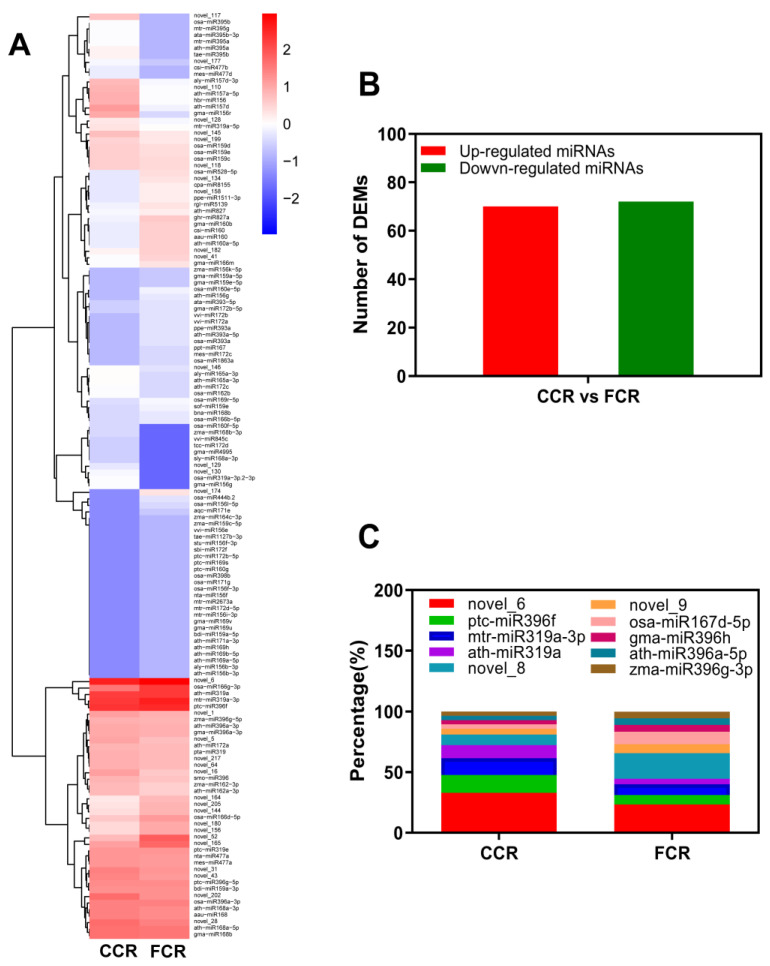
DEMs expression profile in different groups of kernels. (**A**) Analysis of DEMs heatmaps and clusters. (**B**) Different groups’ DEM numbers and characteristics. (**C**) Top 10 expressed miRNAs in each sample. FCC stands for the root samples of the first cropping, and CCR stands for root samples of consecutive cropping.

**Figure 5 plants-12-00943-f005:**
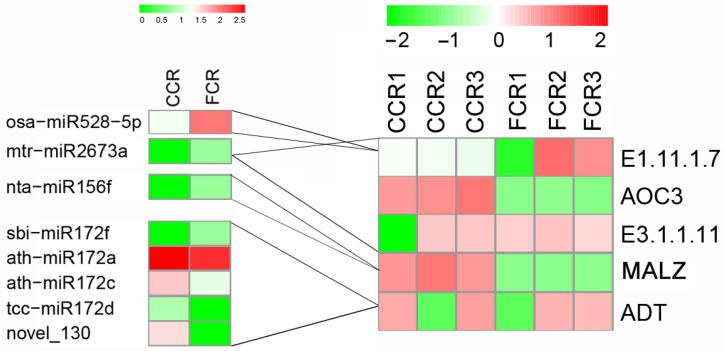
A heatmap displaying differentially expressed miRNAs (DEMs) with their target genes. FCC stands for the root samples of the first cropping, and CCR stands for root samples of consecutive cropping.

**Figure 6 plants-12-00943-f006:**
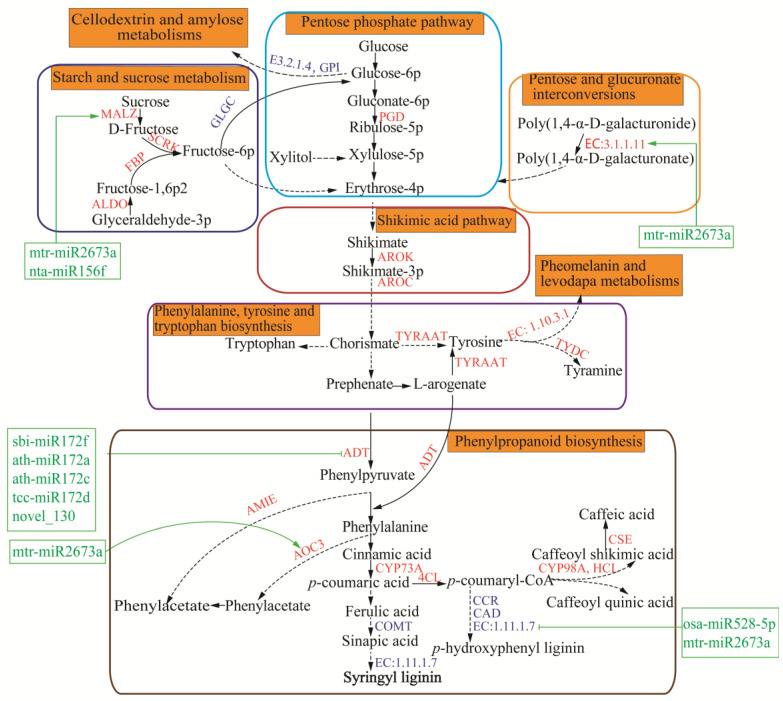
Regulatory network of phenolic acid biosynthesis in the roots of *P. odoratum*. The red letters are up-regulation genes, the blue letters are down-regulation genes, and the green letters are miRNAs. An arrow with a solid frame indicates only one step, while an arrow with a dotted frame arrow indicates more than one step. Black arrows indicate directions for the biosynthesis of phenolic acid.

**Figure 7 plants-12-00943-f007:**
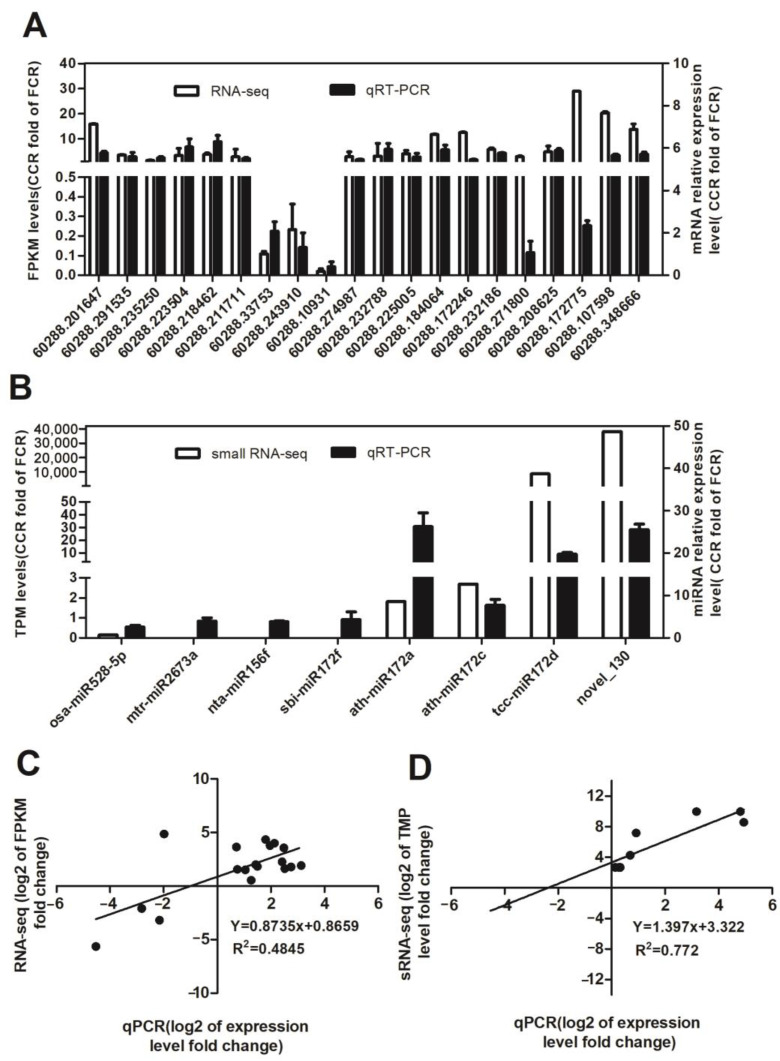
qPCR verifies the quality of transcriptome sequencing and small RNA sequencing in the roots of *P. odoratum*. (**A**) Results from RNA-sequencing on transcript levels and qPCR of 20 unigene genes that regulate the phenolic acid synthesis. (**B**) The small RNA sequencing levels and qPCR results of 8 miRNAs from small RNA-sequencing, which regulates phenolic acid synthesis. The white bar graph represents the normalized result (CCR vs FCR) of the sample in RNA-seq of (**A**) or sRNA-seq (left y-axis) of (**B**), whereas a black bar graph represents the relative expression level (right y-axis). (**C**) The log2 expression ratios from RNA-seq (y-axis) and qPCR (x-axis) were used to generate scatterplots. (**D**) The log2 expression ratios from small RNA-seq (y-axis) and qPCR (x-axis) were used to generate scatterplots. In the left y-axis are the relative gene expression levels analyzed by qPCR (black lines). FCC stands for the root samples of the first cropping, and CCR stands for root samples of consecutive cropping.

## Data Availability

Illumina HiSeq generated RNA-Seq reads and the small RNA sequencing datasets are available in NCBI Sequence Read Archive (SRA) for the Bioproject: PRJNA507291.

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
