# Peer review of "Integrated Analysis of microRNA and RNA-Seq Reveals Phenolic Acid Secretion Metabolism in Continuous Cropping of Polygonatum odoratum"

_plants, 2023, doi:10.3390/plants12040943_

Round 1

Reviewer 1 Report

This manuscript presents the results of the study, which follows the author´s previous analysis (published at BMC Plant Biology 2021, https://doi.org/10.1186/s12870-021-03135-x)

The novelty is further analysis of differentially expressed genes and miRNAs potentially involved in gene expression regulation.

Seems that there was no new soil chemistry analysis done, so it could be solely refered to previous study and shorten this parts also in methods.

My major concern is on the RNAseq analysis of miRNAs, based on the described methodology of RNA isolation and sequencing, as these are mainly design for mRNAs and not for small RNAs which will be under-represented.

The function of miRNA is essentially to pair with potential target mRNA and than direct its cleavage and degradation. Would be usefull to show clearly the targets. How precisely the qPCR was done for such small processed miRNAs?

What about RNA extraction - you have taken roots from field grown plants- this means that there must but quite a lot of microbial, fungal RNA present, how this was assessed?

The assumption of the work is that respective phenylpropanoid biosynthesis pathways genes, and their respective proteins produce given metabolites which are eventually secreted out of the root and get accummulated in the soil. This is missing to be demonstrated, as there is shown expression pattern, but no activities. It could well be that discussed phenolic compounds are not originating from roots, but are results of soil chemistry altered by other root exudates. It would be usefull to show this for example in hydroponic culture. These metabolites might be possible the results of older roots decomposition. This should be discussed.

The statement that phenolic compounds are causing root rot is strange, since many of these substances are actually antimicrobial. How this was shown?

Statements in abstract lines 27-29 should be rephrased, since miRNAs can not it "beneficial, or helpfull", the can just regulate.

Minor but important points:

check on the species names - there all should be in italics, which is not a case.

Gene names, numbers are inconsistent, for example line 232: (genes encoding malz, scrK, ALDO, FBP, SORD, rep, glgC,  etc.) lower and upper letters, should be standardized,

There are some typing errors.

Keywords: I suggest to omitt these (continuous cropping soil; first cropping soil) and place instead soil phenolics, root exudates, root,

Author Response

This manuscript presents the results of the study, which follows the author´s previous analysis (published at BMC Plant Biology 2021, https://doi.org/10.1186/s12870-021-03135-x)

1.The novelty is further analysis of differentially expressed genes and miRNAs potentially involved in gene expression regulation.

Seems that there was no new soil chemistry analysis done, so it could be solely refered to previous study and shorten this parts also in methods.

Response 1: Thanks for your suggestion! Quantification analyses of phenolic acid should been shortened in methods. And we also detected several new chemicals in soil as shown in fig 1.

  1. My major concern is on the RNAseq analysis of miRNAs, based on the described methodology of RNA isolation and sequencing, as these are mainly design for mRNAs and not for small RNAs which will be under-represented.

Response 2: Thanks for your suggestion! We have added “RNA isolation, quantification, and qualification” in the "Materials and methods" section. We used the same batch of RNA for both the RNA-seq and miRNA sequencing analyses.

  1. The function of miRNA is essentially to pair with potential target mRNA and than direct its cleavage and degradation. Would be usefull to show clearly the targets. How precisely the qPCR was done for such small processed miRNAs?

Response 3: Target genes of miRNAs were predicted using the psRNA Target program (http://plantgrn.noble.org/psRNATarget) by Novogene Technology Co., Ltd. (Beijing, China). For qPCR verification of differentially expressed miRNA, we used miRNA mature sequences as templates, and the primers were designed by the stem-loop method. The stem-loop structure not only effectively lengthens the length of miRNA but also avoids binding with other homologous genes in its complementary conformation, which reduces the probability of non-specific amplification. Therefore, it is relatively accurate to verify miRNA by the stem-loop method.

  1. What about RNA extraction - you have taken roots from field grown plants- this means that there must but quite a lot of microbial, fungal RNA present, how this was assessed?

Response 4: After taking the roots of P. odoratum from the field, we cleaned rhizomes with distilled water to remove the soil and some microorganisms cut into pieces, then extracting RNA. However, there is still a small amount of microbial and fungal RNA. However, the transcript sequence obtained by splicing with Trinity was used as a reference sequence for subsequent analysis. Theoretically, these matched reads were all form this species.

5.The assumption of the work is that respective phenylpropanoid biosynthesis pathways genes, and their respective proteins produce given metabolites which are eventually secreted out of the root and get accummulated in the soil. This is missing to be demonstrated, as there is shown expression pattern, but no activities. It could well be that discussed phenolic compounds are not originating from roots, but are results of soil chemistry altered by other root exudates. It would be usefull to show this for example in hydroponic culture. These metabolites might be possible the results of older roots decomposition. This should be discussed. The statement that phenolic compounds are causing root rot is strange, since many of these substances are actually antimicrobial. How this was shown?

Response 5: Thanks for your kind suggestion! We accept your suggestions. We have added the depiction of “originating of phenolic compounds” as follows: These results indicated that the CC system increased phenolic acid accumulation from the rhizosphere soil of P. odoratum. Phenolic acids, as allelopathic substances, are mainly secreted by plant rhizomes but can also be decomposed by old roots and produced by soil microorganisms. (Li et al. Plant phenolics: Recent advances on their biosynthesis, genetics, and ecophysiology, https://doi.org/10.3390/molecules15128933.; Cheynier et al, Plant phenolics: recent advances on their biosynthesis, genetics, and ecophysiology, https://doi.org/10.1016/j.plaphy.2013.05.009. ). In the present study, we speculated that they may be secreted by the rhizomes of P. odoratum, thereby contributing to CMP and replant disease.

6.Statements in abstract lines 27-29 should be rephrased, since miRNAs can not it "beneficial, or helpfull", the can just regulate.

Response 6: Thanks for your suggestion! We have revised the statements in abstract lines 27-29.

Minor but important points:

7. check on the species names - there all should be in italics, which is not a case.

Gene names, numbers are inconsistent, for example line 232: (genes encoding malz, scrK, ALDO, FBP, SORD, rep, glgC,  etc.) lower and upper letters, should be standa                                                    rdized,

There are some typing errors.

Response 7: Thank you for your kind reminder. We have carefully checked species names, gene names, numbers in the manuscript and revised them. All the changes would be seen in the revised manuscript.

8. Keywords: I suggest to omitt these (continuous cropping soil; first cropping soil) and place instead soil phenolics, root exudates, root,

 Response 8: According to the reviewer`s suggestion, We have revised in “Keywords”.

Reviewer 2 Report

In this manuscript from Wang et al., the authors found that Phenolic acids accumulate in soils of continuously crop Polygonatum odoratum, which result in reducing yield and quality of crop. Through performing combined microRNA (miRNA)-seq and RNA-seq analysis in P. odoratum roots in response to consecutive monoculture problem (CPM), the authors found that there were 15,788 differentially expressed genes (DEGs) and 142 differentially expressed miRNA (DEMs) in roots from FC compared with CC plants. They identified 28 DEGs and 8 DEMs that involved in phenolic acid biosynthesis that might play vital roles in phenolic acid secretion from roots of P. odoratum under the CC system.

Overall, the introduction of this manuscript provides sufficient background and include most relevant references. The research design appropriately. The methods were properly described. This manuscript was well written and easy to follow. The following are some minor concerns need to be modified.

1.       Line 17-19, this sentence needs some modification. It causes confusion and it is difficult to understand.

2.       Line 206-216, the description of results in section 2.4 was not consistent with that showed in figure 5. The results in figure 5 need further validation by RT-PCR to show these miRNA and target mRNA candidates are true interaction pairs.

3.       Line 342, “Co nversely” should be “conversely”.

Author Response

  1. Line 17-19, this sentence needs some modification. It causes confusion and it is difficult to understand.

Response 1: Thank you for your suggestion. Line 17-19, this sentence have been modificated as follows: The phenolic acid contents of the first cropping (FC) soil and CC soil were determined by HPLC analysis. The results showed that CC soils contained significantly higher levels of p-coumaric acid, phenylacetate, and caffeic acid than FC soil, except for cinnamic acid and sinapic acid.

  1. Line 206-216, the description of results in section 2.4 was not consistent with that showed in figure 5. The results in figure 5 need further validation by RT-PCR to show these miRNA and target mRNA candidates are true interaction pairs.

Response 2: Thank you for your comment. We quite agree with your opinion. Each gene has many unigenes; for example, ADT and E1.11.1.7 have 8 and 93 unigenes, respectively. It isn't easy to verify by qPCR in the short term. We will pay attention to this in future work. Thanks again.

  1. Line 342, “Co nversely” should be “conversely”.

Response 3: Thank you for your kind reminder. we have revised “Co nversely” to “conversely” in the manuscript.

Round 2

Reviewer 1 Report

Good work, revisions reflecting all rised comments.